# In Silico Identification and Biological Evaluation of Antioxidant Food Components Endowed with Human Carbonic Anhydrase IX and XII Inhibition

**DOI:** 10.3390/antiox9090775

**Published:** 2020-08-21

**Authors:** Giosuè Costa, Annalisa Maruca, Roberta Rocca, Francesca Alessandra Ambrosio, Emanuela Berrino, Fabrizio Carta, Francesco Mesiti, Alessandro Salatino, Delia Lanzillotta, Francesco Trapasso, Anna Artese, Stefano Alcaro, Claudiu T. Supuran

**Affiliations:** 1Dipartimento di Scienze della Salute, Università “Magna Græcia” di Catanzaro, Campus “S. Venuta”, Viale Europa, 88100 Catanzaro, Italy; gcosta@unicz.it (G.C.); maruca@unicz.it (A.M.); ambrosio@unicz.it (F.A.A.); francesco.mesiti@studenti.unicz.it (F.M.); artese@unicz.it (A.A.); 2Net4Science Academic Spin-Off, Università “Magna Græcia” di Catanzaro, Campus “S. Venuta”, Viale Europa, 88100 Catanzaro, Italy; rocca@unicz.it; 3Dipartimento di Medicina Sperimentale e Clinica, Università “Magna Græcia” di Catanzaro, Campus “S. Venuta”, Viale Europa, 88100 Catanzaro, Italy; salatino@unicz.it (A.S.); delialanzillotta@unicz.it (D.L.); trapasso@unicz.it (F.T.); 4Dipartimento NEUROFARBA, Sezione di Scienze Farmaceutiche, Università degli Studi di Firenze, Sesto Fiorentino, 50019 Florence, Italy; emanuela.berrino@unifi.it (E.B.); fabrizio.carta@unifi.it (F.C.); claudiu.supuran@unifi.it (C.T.S.)

**Keywords:** *h*CA IX and XII, dual inhibitors, molecular modeling studies, in vitro assays

## Abstract

The tumor-associated isoenzymes *h*CA IX and *h*CA XII catalyze the hydration of carbon dioxide to bicarbonate and protons. These isoforms are highly overexpressed in many types of cancer, where they contribute to the acidification of the tumor environment, promoting tumor cell invasion and metastasis. In this work, in order to identify novel dual *h*CA IX and XII inhibitors, virtual screening techniques and biological assays were combined. A structure-based virtual screening towards *h*CA IX and XII was performed using a database of approximately 26,000 natural compounds. The best shared *hits* were submitted to a thermodynamic analysis and three promising best *hits* were identified and evaluated in terms of their *h*CA IX and XII inhibitor activity. In vitro biological assays were in line with the theoretical studies and revealed that syringin, lithospermic acid, and (-)-dehydrodiconiferyl alcohol behave as good *h*CA IX and *h*CA XII dual inhibitors.

## 1. Introduction

The Carbonic Anhydrases (CAs, EC 4.2.1.1) are metalloenzymes that reversibly catalyze the hydration of carbon dioxide (CO_2_) to bicarbonate (HCO_3_^−^) and proton (H^+^) ions. These enzymes are grouped into eight distinct and not related genetic families (α, β, γ, δ, ζ, η, θ, and ι-CAs) and are a typical example of biological convergent evolution directed toward the catalysis of essential biochemical processes [1,2,3]. The human CAs (*h*CAs) belong to the α-class and exist in 15 different isoforms, which differ for tissue distribution, subcellular localization, and catalytic activity. In particular, *h*CA I-III, *h*CA VII, and *h*CA XIII are cytosolic isozymes; *h*CA VA and VB are located in the mitochondria; and *h*CA IV, *h*CA IX, *h*CAXII, and *h*CA XIV are membrane-bound isozymes [4]. These enzymes are distributed in human tissues and organs, where they are involved in critical physiological process, including pH regulation, electrolyte secretion, respiration, bone resorption, among others [5]. Therefore, abnormal levels and/or activities of these enzymes usually are associated with several diseases, e.g., glaucoma, neurological disorder, osteoporosis, and metabolic disorders [6].

Recently, two specific *h*CA isoforms, namely *h*CA IX and *h*CA XII, gained great attention [7]. They usually are referred to as the “tumor associate isoenzymes” in consideration of the specific localization and activity within hypoxic tumor tissues. In detail, *h*CA IX is ectopically induced and highly overexpressed in many solid tumors, including brain, breast, bladder, pancreas, and T-cell lymphomas [8]. Furthermore, it was observed that *h*CA IX is closely overexpressed in response to hypoxia in cancer cells, whereas *h*CA XII isoform is also upregulated in many tumor types, however, its activity and expression are also profuse in normal tissues [9,10]. Several pieces of evidence demonstrated that *h*CA IX and *h*CA XII inhibitors affect the pH of the tumor microenvironment, reducing tumor cell survival and proliferation [11,12]. Due to their implication in tumorigenicity and cancer metastasis, *h*CA IX and *h*CA XII isoforms have been widely investigated, and several *h*CAs inhibitors were developed as promising anticancer agents. Encouraging in vitro and in vivo studies have shown that the inhibition of *h*CA IX and *h*CA XII decreases growth, proliferation, and metastatic potential of different cancers [13,14]. In this respect, the sulfonamide (and their structural-related bioisosteres, sulfamates and sulfamides) and the coumarins/thiocoumarins-based *h*CA IX and *h*CA XII inhibitors showed promising results [15,16,17,18]. In parallel, natural occurring compounds, especially phenol derivatives, were figured out to have significant antioxidant activities and remarkable *h*CAs inhibitory capacity [19,20,21]. Guaiacol, a plant isolated compound, and several catechol derivatives effectively inhibited *h*CA IX and *h*CA XII with K_i_ at low micromolar range [21].

Nowadays, cancer is ranked as the second cause of death worldwide, therefore, the development of new anticancer therapies is an urgent need [22,23]. In particular, the enhanced reducing conditions, which usually account for a pO_2_ less than 1%, represent the main feature of hypoxic tumors environment. Among the drugs which make use of such conditions (i.e., better referred as hypoxia activable prodrugs) [24,25,26], phenols and catechols are the major ones whose antioxidant activity is widely reported and documented [27,28]. Furthermore, the design of new molecules with antioxidant properties, coupled to a multi-target profile, could be a useful approach to face oncogenesis and cancer progression [29,30,31,32]. Moreover, in the development of *h*CAs inhibitors, special attention should be given to their selective profile. In fact, several drug interactions and side effects have been linked to non-selective *h*CAs inhibition, thus, a selective profile over the appropriate isoforms is mandatory [7,33]. Under this light, selective *h*CA IX and *h*CA XII inhibitors could be considered as potential anticancer drugs [18].

In the drug discovery process, the application of computational methods has demonstrated to be important for the identification of new *hit* compounds. The in silico studies are able to speed up the identification of bioactive compounds, thus, reducing the cost and time of research-associated activities [34]. In particular, structure-based virtual screening (SBVS) consists of a computational tool helpful to identify novel bioactive ligands towards a selective target(s), exploiting the three-dimensional (3D) structures of the biological target, either protein or nucleic acid [35,36,37,38,39,40,41]. Many natural products and/or food constituent molecules have been considered in the treatment of serious diseases, including cancer [42,43]. FooDB [44] is the most comprehensive online database providing information about food constituent molecules, their chemical structures, and concentrations in various foods. Within this framework and in the continuous research of potent and selective *h*CA IX and *h*CA XII inhibitors, we performed a SBVS using a database of natural occurring compounds. The FooDB database was used to build up a chemical library of natural occurring compounds virtually screened towards several *h*CAs. The most promising compounds, based on their theoretical binding energy, were further submitted to in vitro assays to point out new promising *h*CA IX and *h*CA XII inhibitors. The experimental results confirmed the computational predictions, providing the rationale behind the ligands biological activity and selectivity.

## 2. Materials and Methods

### 2.1. Molecular Modeling Studies

Starting from the three dimensional structure of the human Carbonic Anhydrase IX in complex with 5-(1-naphthalen-1-yl-1,2,3-triazol-4-yl)thiophene-2-sulfonamide and the crystal structure of the human Carbonic Anhydrase isozyme XII with 2,3,5,6-tetrafluoro-4-(propylthio)benzenesulfonamide, deposited in the Protein Data Bank (PDB) with the PDB codes 5FL4 [45] and 5MSA [46], respectively, our molecular modeling simulations were carried out.

With the aim to evaluate the reliability of our molecular recognition approach, we performed redocking calculations by using the Glide Standard Protocol (SP) algorithm, which was able to reproduce the experimentally determined binding modes, obtaining Root Mean Square Deviation (RMSD) values, calculated between the best docking pose and the ligand co-crystallized into the *h*CA IX and XII catalytic binding site, equal to 0.545 and 1.059 Å for *h*CA IX and *h*CA XII, respectively.

The receptor structures were energy optimized by means of the Protein Preparation Wizard tool implemented in Maestro, using OLPS_2005 as the force field. Residual crystallographic buffer components and water molecules were removed, hydrogen atoms were added, missing side chains were built using the Prime module, and side chains protonation states at pH 7.4 were assigned [47,48].

For the Virtual Screening (VS) studies, the FooDB database, containing 26,680 compounds, was used [44]. The library was prepared by means of the LigPrep tool [49], hydrogens were added, salts were removed, ionization states were calculated using Ionizer at pH 7.4, and then, all the compounds were submitted to energy minimization calculations, using OPLS_2005 as the force field, thus, obtaining 25,584 compounds. Glide v. 6.7 SP algorithm was used to perform VS [50] and 10 poses for ligands were generated. 

With the purpose to select the scored compounds according to their Glide score (G-score) value, we performed molecular docking studies of already approved and investigational *h*CA isoforms IX (**2**–**5**) and XII (**1**–**5**) inhibitors (Appendix A), such as acetazolamide, zonisamide, and ellagic acid, and we obtained a consensus value to be applied as a filter (Appendix A). Starting from the best shared *hits* that resulted from VS simulations, a post-docking energy minimization was applied using the eMBrAcE tool developed by Schrödinger (MacroModel v10.8) [51,52] and the binding energies (ΔE) between ligands and receptors were calculated. Each complex was subjected to energy minimization in the implicit solvent, using OPLS_2005 as the force field and the conjugate gradient protocol. The average ΔE values of the already approved and investigational *h*CA isoforms IX and XII inhibitors, respectively, were used to further filter the obtained *hits*.

Finally, the shared compounds between both *h*CA isoforms IX and *h*CA XII were investigated by visual inspection. Then, considering their commercial availability, we purchased the 3 best *hits*, which were subjected to in vitro assays.

### 2.2. Carbonic Anhydrase Inhibition Assay

The CA-catalyzed CO_2_ hydration activity of the compounds reported was assayed by means of the Applied Photophysics stopped-flow instrument [53]. Phenol red was used as a pH indicator at a concentration of 0.2 mM, the instrument was set at working maximum absorbance maximum of 557 nm, the solutions were buffered at pH 7.5 with 20 mM HEPES, and 20 mM Na_2_SO_4_ was added for keeping constant the ionic strength, to follow the initial rates of the enzyme-catalyzed CO_2_ hydration reaction for a period of 10–100 s. The CO_2_ concentrations for the determination of the kinetic parameters and inhibition constants ranged from 1.7 to 17 mM. For each potential inhibitor, at least six traces of the initial 5–10% of the reaction have been used for determining the initial rate.

The uncatalyzed rates were determined in the same experimental manner and subtracted from the total observed values. Stock solutions of the potential inhibitor (0.1 mM) were prepared by using distilled deionized water, and the dilutions with the assay buffer were prepared up to 0.01 nM. The solutions containing the inhibitor and the enzyme were preincubated for 30 min at r.t. prior to performing the assay, in order to form the E−I adduct. The inhibition constants (K_IS_) were obtained by the nonlinear least squares methods using PRISM 3 and the Cheng−Prusoff equation, as reported earlier [54,55], and represent the mean from three different experimental determinations. All CA isoforms herein reported were recombinant and obtained *in-house* [54,55].

### 2.3. Inhibition Growth Assay

The human colorectal cancer cells Caco-2 were maintained in Dulbecco’s Modified Eagle Medium (DMEM, Sigma-Aldrich, St. Louis, MO, USA) integrated with 10% of FBS (Fetal Bovine Serum, Sigma-Aldrich, St. Louis, MO, USA) and 1% P/S (Penicillin, Streptomycin, Sigma-Aldrich, St. Louis, MO, USA). Cells’ culture conditions were 5% CO_2_ incubator at 37 °C. Cell viability was assessed through an MTT assay: 1 × 10^4^ cells/well were plated in a 96-well microplate and, two hours later, treated with **6**, **12**, and **14** (the latter dissolved in DMSO) at 10 µM concentration. After 24, 48, and 72 h, 20 µL of 3-(4,5-dimethylthiazol-2-yl)-2,5 diphenyl tetrazolium (MTT detection reagent, Sigma-Aldrich, St. Louis, MO, USA) was added on the plate at the final concentration of 5mg/mL, and incubated for two more hours [56]. Culture agent was removed and 200 µL of isopropanol (0.04 M HCl) were added to solubilizeformazan crystals. Absorbance was measured at 560 nm by the Victor 3 reader.

## 3. Results

### 3.1. Structure-Based Virtual Screening (SBVS)

In this study, by using a SBVS approach, about 26 thousand food components were screened against both *h*CA IX and XII isoforms. Starting from the active set (Appendix A), we applied a *cut-off* to filter the obtained *hits*, thus, considering for *h*CA IX and XII, the average Glide SP scores of −5.11 and −5.38 kcal/mol, respectively.

This first filter led to selecting 7364 and 6552 *hits* from *h*CA IX and XII, respectively. Afterwards, we calculated the binding energies of the 4430 shared *hits* in complex with both receptors, with the aim to investigate their thermodynamic profile. In order to select the best promising *hits* with a potential dual activity, we applied the same protocol also for the active set (**1**–**5**) (Appendix A). Therefore, their average ΔE values, equal to −21.92 and −28.19 kcal/mol for *h*CA IX and XII, respectively, were used as the *cut-off*, globally leading to 988 *hits*, characterized by a good theoretical dual activity.

Finally, after a careful visual inspection analysis of the best poses, nine *hits* (**6–14**) were selected (Appendix A). Structurally, all the selected *hits* showed hydroxides, phenols, and carboxylic acids as recurring chemical scaffolds, confirming the essential role of these moieties in the molecular recognition of both *h*CA IX and XII isoforms (Appendix A). Unfortunately, the evaluation of their commercial availability revealed that only three compounds (**6**, **12**, and **14**) could be purchased and submitted for biological assays. Among them, compound **12** was found to have a good inhibitory activity on *h*CA XII [28], thus, we included it in the biological screening also on the IX isoform, to further validate our protocol aimed at discovering new dual inhibitors for both *h*CAs involved in tumors pathogenesis.

All the VS steps are summarized in Figure 1.

### 3.2. Carbonic Anhydrase Inhibition Assay

The selected compounds **6**, **12**, and **14** were evaluated in vitro for their inhibition potencyagainst the most relevant *h*CAs such as the cytosolic and widely expressed *h*CAs, I and II, and the tumor-associated *h*CAs, IX and XII. The obtained data reported below in Table 1 were all compared to the standard *h*CA inhibitor acetazolamide (**AAZ**).

Overall, the compounds resulted ineffective inhibitors against the widely expressed *h*CAs I and II with K_I_s >100 µM. Interesting results were obtained for the remaining isoforms:(i)As for compound **14**, the data reported in Table 1 clearly showed the tumor-associated *h*CA XII isoform was 3.5-fold more potently inhibited when compared to the *h*CA IX (K_I_s of 0.092 and 0.32 µM, respectively) with a selectivity index (SI; K_I_
*h*CA IX/K_I_
*h*CA XII) of 3.5.(ii)The same kinetic profile was also recovered for **6**, although enhanced enzymatic SIs were observed. The tumor-associated *h*CA XII resulted inhibited 27.0-fold more potently when compared to the IX (K_I_s of 0.096 and 2.59 µM, respectively).(iii)Compound **12** was already investigated as a potential inhibitor *h*CA on the widely expressed *h*CAs, I and II, and on the tumor-associated IX and XII from some authors of this manuscript and the data reported in Table 1 were in good agreement [28]. It is worth considering that an impressive *h*CA XII selective inhibition was obtained from such a substance, thus being the most potent and selective against such a tumor-associated isoform (K_I_s of 0.31 and 0.0048 µM for the *h*CA IX and XII, respectively).

The kinetic data reported here clearly showed compounds **6**, **12**, and **14** being effective inhibitors of the tumor-associated *h*CA IX and XII and in particular, the latter was preferentially inhibited with K_I_ values in the low micromolar range. Among the substances tested, compound **12** was a particularly potent inhibitor of the *h*CA XII, having a K_I_ value close to the reference drug of the sulfonamide type AAZ (K_I_s of 0.0048 and 0.0057 µM, respectively).

### 3.3. Docking Pose and Thermodynamic Analysis of the Best Hits

By evaluating their best docking poses, all three *hits* were well recognized in the catalytic binding site of both *h*CA IX and XII, as shown in Figure 2. By using the Maestro graphical interface [57] contact analysis, we observed all compounds were able to strongly interact with the binding pocket residues of both tumor-associated *h*CAs by means of hydrogen bond (H-bond), salt bridges, electrostatic, and cation interactions. Moreover, the better affinity of all three *hits* towards *h*CA XII could be rationalized by the higher number of good contacts and H-bonds with respect to the *h*CA IX isoform (Table 2).

Regarding **12**’s best pose, we found its two carboxyl groups involved in two salt bridge interactions with the side chain of Lys69 and Lys3 in the *h*CA XII binding pocket (Figure 2e). The same groups engaged different H-bonds with Lys3, Trp4, Lys69, and Gln89, thus, further stabilizing the complex. Moreover, the pyrocatechol ring established a cation interaction and an H-bond with the side chains of Lys3 and Asn94, respectively, while the pyrocatechol linked to dihydrofuran interacted with Thr198 by means of two H-bonds. Conversely, as shown in Figure 2b, the pyrocatechol linked to the dihydrofuran moiety established four H-bonds with the side chains of Asn66 and Arg64; meanwhile, the other pyrocatechol ring was involved in a stacking interaction and one H-bond with the side chain of His94 and Tyr11, respectively. With respect to *h*CA XII, we highlighted that **12** formed an H-bond network among its two carboxyl groups and the residues Trp9, Hie68, and Gln92.

Regarding compound **6**, the alcohol groups were involved in the most important interactions in both isoforms pockets (Figure 2a–d). Specifically, in *h*CA IX, the allyl alcohol of compound **6** engaged an H-bond with the side chain of Thr200, while the primary alcohol interacted with the side chain of Trp9 and the backbone of Pro202 (Figure 2a). Conversely, in *h*CA XII, the same groups were involved in two H-bonds with Thr198 and Asn64. The two additional H-bonds, found between the phenolic moiety of compound **6** and the side chains of Lys3 and Trp9, further increased the affinity of this compound towards the XII isoform, as confirmed by the biological assays (Figure 2d). Finally, as reported in Table 2, compound **6** was able to establish several productive interactions with both the *h*CA isoforms.

By analyzing the binding pose of compound **14**, we observed that the sugar moiety and the allylic alcohol were implicated in pivotal interactions. In detail, we found that the sugar moiety established two H-bonds with *h*CA IX, Gln71 and Gln92; meanwhile, the same portion was able to form three H-bonds with Lys69, Thr88, and Val119 residues of *h*CA XII. Finally, the allylic alcohol was involved in an H-bond with Thr200 and Thr198 of the IX and XII isoforms, respectively.

The detailed evaluation of compounds **6**, **12**, and **14** thermodynamic profile against *h*CA IX and XII isoforms showed the eMBrAcE protocol able to well predict their good dual activity, as confirmed by the biological results. By investigating the single contributions of the ΔE energy components, we observed the electrostatic term as the driven force in the binding process for both targets (Table 3).

### 3.4. Cell Viability Assay

In order to investigate the effects of compounds **6**, **12**, and **14**, we carried out an MTT assay on Caco-2 cancer cells. Cell viability was assessed at 24, 48, and 72 h after the cells’ treatment.

No significant effects were assessed 24 h after treatment (data not shown).

Interestingly, the most significant effects were observed 72 h after treatment with all tested compounds, ranging from nearly 70% of cell viability (**6**) to approximatively 50% of cell viability (**12** and **14**) (Figure 3).

Considering the aforementioned results, those compounds can have a potential anticancer activity, thus, encouraging their further development.

## 4. Discussion

The molecular modeling studies presented evidence of the presence of the phenolic moiety in all best *hits*. Such a scaffold is known to be incorporated in several antioxidant compounds and, more recently, was found to have a key role in inhibiting *h*CA I and II [58].

Compounds **6**, **12**, and **14** are food constituents, thus, confirming the strong importance of natural components as a source of lead and bioactive compounds, able to affect several biological processes in our organism. In fact, many studies in the literature have demonstrated that natural products with anti-inflammatory, antimicrobial, and antioxidant activity could be useful for multifactorial diseases [42,43].

Among the three *hits*, compound **6** is known as (-)-dehydrodiconiferyl alcohol, and belongs to the class of organic compounds known as 2-arylbenzofuran flavonoids. It has been detected, but not quantified, in coffee and coffee products and green vegetables. In the literature, it was found to exert anti-*Helicobacter pylori* [59], antiadipogenic [60], and antioxidant effects [61]. Recently, Lee et al. reported that (-)-dehydrodiconiferyl alcohol suppresses the p38 MAPK and NF-κB signaling pathways in RAW 264.7 cells and acts as an estrogen receptor agonist [62]. Moreover, it exerts anti-inflammatory activity by regulating key molecules involved in inflammation and oxidative stress, such as pro-inflammatory cytokines (TNF-α, IL-6 and IL-1β) and mediators (iNOS, COX-2, and ROS) [63].

As with compound **6**, the lithospermic acid **12** is a member of the class of 2-arylbenzofuran flavonoids and it is also known as lithospermate. Our approach highlighted it as the most interesting dual inhibitor of both *h*CAs, in line with a previous work related to its *h*CA XII [28] proven activity. It can be found in common thyme and peppermint, which makes lithospermic acid a potential biomarker for the consumption of these food products, and it showed a good and well-known antioxidant activity [64,65].

Finally, compound **14** is syringin, also known as eleutheroside b or β-terpineol. It belongs to phenolic glycosides, which are organic compounds containing a phenolic structure attached to a glycosyl moiety. Syringin was first isolated from the bark of lilac (*Syringa vulgaris*) by Meillet in 1841. Moreover, it can be found in caraway, fennel, and lemon, which makes syringin a potential biomarker for the consumption of these food products. Several pharmacological actions of syringin include plasma glucose reduction, antioxidation, anticancer activity, antidepressant effect, and immunomodulation [66,67].

*In vitro* results demonstrated that all the three *hits* are able to inhibit both *h*CA IX and XII isoforms in the micromolar range, although with a preference towards the XII target, thus, confirming their potential anticancer activity.

## 5. Conclusions

In this work, in silico and experimental techniques were combined to identify natural bioactive compounds present in food, such as coffee, green vegetables, common thyme, peppermint, caraway, fennel, and lemon, and are endowed with inhibition properties against both *h*CA IX and XII isoforms. In particular, we proposed (-)-dehydrodiconiferyl alcohol, lithospermic acid, and syringin as good *h*CA IX and *h*CA XII dual inhibitors with a potential anticancer activity, thus, encouraging their further development. Specifically, lithospermic acid and syringin, with K_I_ values in the low micromolar range, showed a better cytotoxic effect seventy-two hours after treatment than (-)-dehydrodiconiferyl alcohol. Interestingly, all the best *hits* are characterized by a phenol moiety, which has recently aroused considerable interest due to its potential antioxidant effects on human health.

## Figures and Tables

**Figure 1 antioxidants-09-00775-f001:**
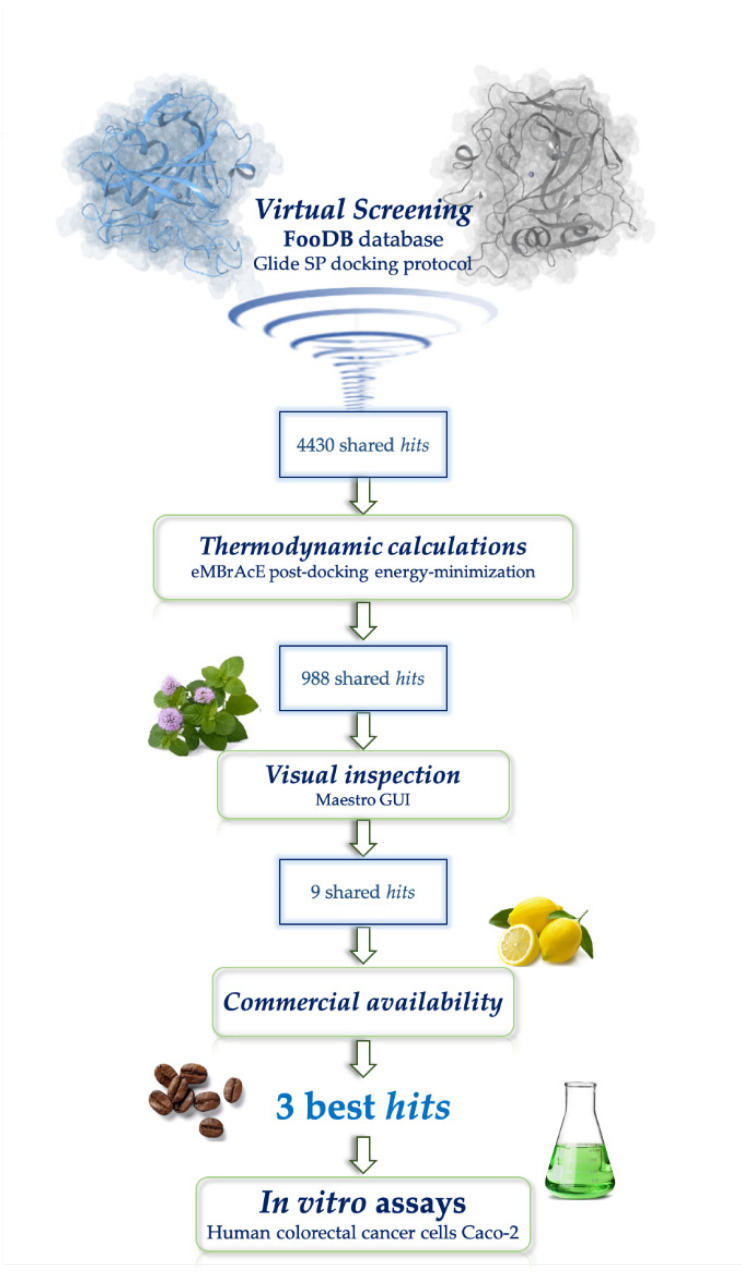
Representation of the SBVSworkflow.

**Figure 2 antioxidants-09-00775-f002:**
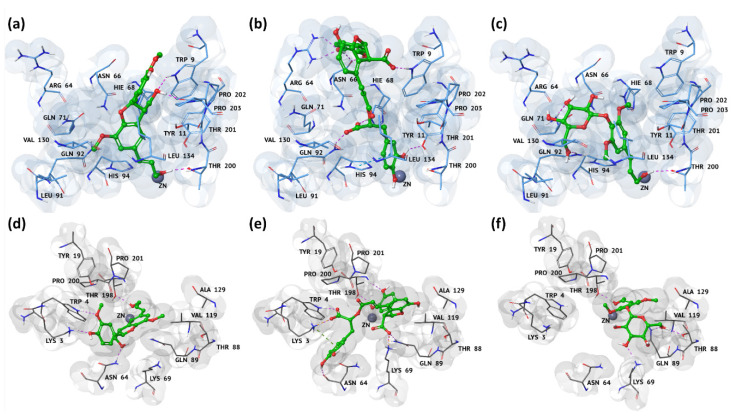
Three-dimensional representations of the binding mode of (**a**–**d**) **6**, (**b**–**e**) **12**, and (**c**–**f**) **14**’s best *hits* against *h*CA IX and XII binding pockets, respectively. The ligands are depicted as green carbon sticks, while *h*CA IX and XII are shown as blue and gray transparent cartoon, respectively. The zinc cation is represented as a violet sphere; the enzyme residues, involved in crucial contacts with the compounds, are reported as blue and gray carbon sticks, respectively, for *h*CA IX and XII isoforms. Hydrogen bonds, salt bridges, cations, and stacking interactions are reported, respectively, as dashed pink, orange, green, and light-blue lines. These binding modes derived from the molecular mechanics energy minimization performed by means of the eMBrAcE tool.

**Figure 3 antioxidants-09-00775-f003:**
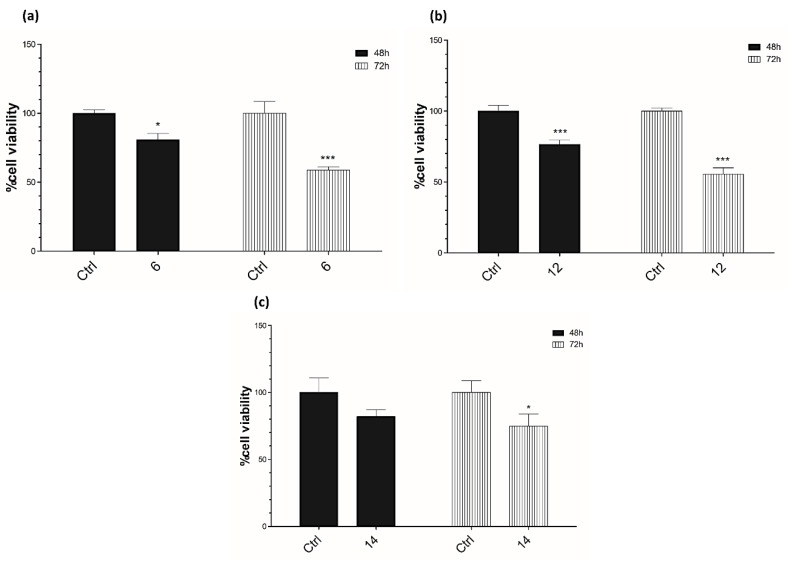
Effects of **6** (**a**), **12** (**b**), and **14** (**c**) on Caco-2 cell viability. A total of 1 × 10^3^ cells were plated, in triplicate, in 96-well plates and treated with compounds at the concentration of 10 μM for 48 and 72 h; cells treated with DMSO were used as control. Cell viability was measured performing an MTT assay and expressed as a percentage of control, analyzed by ANOVA (* *p* < 0.05; ** *p* < 0.005; *** *p* < 0.0005); each column represents the mean ± SD of three different wells.

**Table 1 antioxidants-09-00775-t001:** Inhibition data of *h*CA I, II, IX, and XII with **6**, **12**, **14**, and the standard sulfonamide inhibitor acetazolamide (**AAZ**) by the stopped-flow CO_2_ hydrase assay.

		K_I_ (µM) *			
2D Chemical Structure	*Hit*	*h*CA I	*h*CA II	*h*CA IX	*h*CA XII
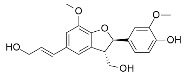	**6**	>100	>100	2.59	0.096
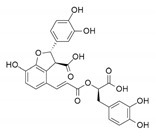	**12**	>100 [28]	>100 [28]	0.31	0.0048 [28]
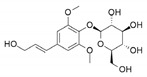	**14**	>100	>100	0.32	0.092
	**AAZ**	0.250	0.012	0.026	0.0057

* Mean ofthree different assays by a stopped-flow technique (errors were in the range of ±5–10% of the reported values).

**Table 2 antioxidants-09-00775-t002:** Hydrogen bonds (HB), salt bridges (SB), good contacts (GC), stacking (S), and cation (C) interactions established by the three best *hits* with the *h*CA IX and XII receptor targets.

	*h*CA IX					*h*CA XII				
*Hit*	HB	SB	GC	S	C	HB	SB	GC	S	C
**6**	3	0	179	0	0	4	0	222	0	0
**12**	8	0	167	1	0	7	2	291	0	1
**14**	3	0	224	0	0	4	0	222	0	0

**Table 3 antioxidants-09-00775-t003:** eMBrAcE ΔE values and their single contributions, expressed as electrostatic, van der Waals, and solvation components (ΔE_Elec_, ΔE_vdW_, and ΔE_Solv_), calculated for the three best *hits* complexed with *h*CA IX and XII receptors. All thermodynamic values are reported in kcal/mol.

		*h*CA IX				*h*CA XII			
*Hit*	FooDB ID	ΔE	ΔE_Elec_	ΔE_vdW_	ΔE_Solv_	ΔE	ΔE_Elec_	ΔE_vdW_	ΔE_Solv_
**6**	FDB021188	−37.89	−71.80	−17.28	46.18	−43.75	−110.89	−25.64	81.26
**12**	FDB006174	−23.42	−59.74	−12.72	37.79	−31.83	−227.57	−28.01	206.77
**14**	FDB011657	−31.22	−36.62	−22.44	26.06	−43.81	−105.69	−22.33	73.05

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
