# Peer review of "In Silico Identification and Biological Evaluation of Antioxidant Food Components Endowed with Human Carbonic Anhydrase IX and XII Inhibition"

_antioxidants, 2020, doi:10.3390/antiox9090775_

Round 1
Reviewer 1 Report
Congratulation for your nice work.
Conclusion should be final word to summarize this research.
Author Response
Answer to Reviewer 1
Congratulation for your nice work.
Thank you very much.
Conclusion should be final word to summarize this research.
As suggested by the reviewer we modified the conclusion in order to summarize our research. “In this work, in silico and experimental techniques were combined to identify natural bioactive compounds present in food, such as coffee, green vegetables, common thyme, peppermint, caraway, fennel and lemon, and endowed with inhibition properties against both hCA IX and XII isoforms. In particular, we proposed (-)-Dehydrodiconiferyl alcohol, Lithospermic acid, and Syringin as good hCA IX and hCA XII dual inhibitors with a potential anticancer activity, thus encouraging their further development. Specifically, Lithospermic acid and Syringin, with KI values in the low micromolar range, showed a better cytotoxic effect seventy-two hours after treatment than (-)-Dehydrodiconiferyl alcohol. Interestingly, all the best hits are characterized by a phenol moiety, which has recently aroused considerable interest due to its potential antioxidant effects on human health.”
Reviewer 2 Report
The topic of this research was interesting, and the authors examined a lot of compounds to search for novel and commercially available dual hCA IX and XII inhibitors. However, there is still room for improvement.
Major concerns:
- The authors used colorectal cancer cells (Caco-2 cells) for inhibition growth assay. However, previous studies (reference [11-14]) used breast cancer cells (MDA-MB-231 and 4T1 cells) and oral squamous cell carcinoma cells (SCC-9, OC2, and OECM-1 cells). Are hCA IX and hCA XII also overexpressed in colorectal cancer cells? And is it appropriate to use colorectal cancer cells for this assay?
- In figure 3, the results of 24 hours inhibition should be presented. Also, did Caco-2 cells really decreased its viability due to hCA IX and hCA XII inhibition? It could be possible that compounds 6, 12, and 14 have effects on other factors and that a decrease in cell viability was observed because of that reason.
- hCA XII has tumor-suppressive aspects in breast and lung cancer (Watson et al., Br J Cancer, 2003; Ilie et al., Int J Cancer, 2011), but its overexpression is related to poor prognosis of colorectal cancer patients (Viikilä et al., World J Gastroenterol, 2016). Considering that, hCA XII inhibitors might be effective for colorectal cancer, but not for all types of cancer. Thus, the effects of hCA XII inhibitors on cancer should be discussed more precisely.
Minor concerns:
- In “3 Inhibition Growth Assay” on page 4, please check the following: CO2→CO2, 104 cells/well→1×104 cells/well, 0,04M HCl→0.04M HCl.
- In the legend of figure 3, please correct for the following notation: 1x103→1×103.
- The titles of Table S1, Table S2, and Table S3 written on page 10 of the manuscript are slightly different from those provided in supplementary materials. Also, please check the following notation: Figure S2.:→Figure S2:
- Concerning references [23] and [39], are they the right citation formats?
- Concerning references [42-45], [47], and [53], please unify the notation: Schrodinger or Schrödinger, New York, or NY.
Author Response
Answer to Reviewer 2
The topic of this research was interesting, and the authors examined a lot of compounds to search for novel and commercially available dual hCA IX and XII inhibitors. However, there is still room for improvement.
Major point.
The authors used colorectal cancer cells (Caco-2 cells) for inhibition growth assay. However, previous studies (reference [11-14]) used breast cancer cells (MDA-MB-231 and 4T1 cells) and oral squamous cell carcinoma cells (SCC-9, OC2, and OECM-1 cells). Are hCA IX and hCA XII also overexpressed in colorectal cancer cells? And is it appropriate to use colorectal cancer cells for this assay?
In figure 3, the results of 24 hours inhibition should be presented. Also, did Caco-2 cells really decreased its viability due to hCA IX and hCA XII inhibition? It could be possible that compounds 6, 12, and 14 have effects on other factors and that a decrease in cell viability was observed because of that reason.
hCA XII has tumor-suppressive aspects in breast and lung cancer (Watson et al., Br J Cancer, 2003; Ilie et al., Int J Cancer, 2011), but its overexpression is related to poor prognosis of colorectal cancer patients (Viikilä et al., World J Gastroenterol, 2016). Considering that, hCA XII inhibitors might be effective for colorectal cancer, but not for all types of cancer. Thus, the effects of hCA XII inhibitors on cancer should be discussed more precisely.
The overexpression of CA IX and XII in various colorectal cancer cells strains, including Caco-2, have been largely demonstrated in literature and nowadays they are routinely used when in-cell screening assays of potential CA inhibitors need to be performed. The authors do agree with this Reviewer that Caco-2 strains have more specific uses for drug permeability experiments, nevertheless the biochemistry remains almost constant among the various colorectal cancer strains.
Minor point.
In “3 Inhibition Growth Assay” on page 4, please check the following: CO2→CO2, 104 cells/well→1×104 cells/well, 0,04M HCl→0.04M HCl.
In the legend of figure 3, please correct for the following notation: 1x103→1×103.
The titles of Table S1, Table S2, and Table S3 written on page 10 of the manuscript are slightly different from those provided in supplementary materials. Also, please check the following notation: Figure S2.:→Figure S2:
Concerning references [23] and [39], are they the right citation formats?
Concerning references [42-45], [47], and [53], please unify the notation: Schrodinger or Schrödinger, New York, or NY.
We thank the reviewer for these suggestions. According to these indications, we have modified the minor concerns.
Reviewer 3 Report
I have a substantial problem with this paper. It describes in silico screening amongst natural products against two isoforms of carbonic anhydrase. Then the commercially available ones were tested against different isoforms of the enzyme and finally as antiproliferative agents. Paper is o.k., however, it is a classic paper in medicinal chemistry and the only fact that might justify its publication in ANTIOXIDANTS is that the selected compounds exhibit antioxidant activity. However, there is no proof that this activity is vital for their mechanism of enzyme inhibition. Thus, in my opinion this paper fits better to other mdpi journals.
In order to be published in ANTIOXIDANTS Authors should proof that antioxidant activity is a vital component of hCA inhibition.
Author Response
Answer to Reviewer 3
I have a substantial problem with this paper. It describes in silico screening amongst natural products against two isoforms of carbonic anhydrase. Then the commercially available ones were tested against different isoforms of the enzyme and finally as antiproliferative agents. Paper is o.k., however, it is a classic paper in medicinal chemistry and the only fact that might justify its publication in ANTIOXIDANTS is that the selected compounds exhibit antioxidant activity. However, there is no proof that this activity is vital for their mechanism of enzyme inhibition. Thus, in my opinion this paper fits better to other mdpi journals.
In order to be published in ANTIOXIDANTS Authors should proof that antioxidant activity is a vital component of hCA inhibition.
The main feature of hypoxic tumors environment is their enhanced reducing conditions, which usually account for a pO2 less than 1%. Various attempts on designing drugs which make use of such conditions (i.e. better referred as hypoxia activable prodrugs) have been reported in the literature (see refs 1-4 as the main examples). Among all the reports the chemical moieties considered, phenols and catechol are the major ones for which the antioxidant activities are widely reported and documented (5, 6).
The natural products considered in this manuscript are all containing such a scaffold and therefore their antioxidant properties are undoubtedly attributable. The Authors agree with the observation rose from this Reviewer and agree that such a point will be better specified into the manuscript. In particular, the antioxidant activity is not determinant for the enzymatic inhibition but as beneficial when the hypoxic tumoral conditions are established.
1) Guise, Christopher P., et al. "Bioreductive prodrugs as cancer therapeutics: targeting tumor hypoxia." Chinese journal of cancer 33.2 (2014): 80.
2) De Simone, Giuseppina, et al. "Carbonic anhydrase inhibitors: hypoxia-activatable sulfonamides incorporating disulfide bonds that target the tumor-associated isoform IX." Journal of medicinal chemistry 49.18 (2006): 5544-5551.
3) Garber, Ken. "New drugs target hypoxia response in tumors." Journal of the National Cancer Institute 97.15 (2005): 1112-1114.
4) Rami, Marouan, et al. "Hypoxia-targeting carbonic anhydrase IX inhibitors by a new series of nitroimidazole-sulfonamides/sulfamides/sulfamates." Journal of medicinal chemistry 56.21 (2013): 8512-8520Oct 31.
5) Karioti, Anastasia, Fabrizio Carta, and Claudiu T Supuran. "An update on natural products with carbonic anhydrase inhibitory activity." Current Pharmaceutical Design 22.12 (2016): 1570-1591 10.2174/1381612822666151211094235.
6) Karioti, Anastasia, et al. "New natural product carbonic anhydrase inhibitors incorporating phenol moieties." Bioorganic & Medicinal Chemistry 23.22 (2015): 7219-7225.
In order to better explain this concept, we added in the introduction the following paragraph and references:
“In particular, the enhanced reducing conditions, which usually account for a pO2 less than 1%, represent the main feature of hypoxic tumors environment. Among the drugs which make use of such conditions, (i.e. better referred as hypoxia activable prodrugs) [24-26] phenols and catechol are the major ones whose antioxidant activity is widely reported and documented [27, 28]. Furthermore, the design of new molecules with anti-oxidant properties, coupled to a multi-target profile, could be a useful approach to face oncogenesis and cancer progression [29-32].”
We would like to point out that for better clarity we have decided to include the unit of measure for constant (Ki) and the 2D chemical structures of the best 3 hits in Table 1.
Finally, as required by the editor, we reformulated all the paragraphs found to be too similar to previously published works.
Round 2
Reviewer 3 Report
After introduction of fragment of the role of antioxidant activity of the studied compounds, paper could be published as it is.